# Remote Radical 1,3-, 1,4-, 1,5-, 1,6- and 1,7-Difunctionalization Reactions [note 1]

**DOI:** 10.3390/molecules28073027

**Published:** 2023-03-28

**Authors:** Xiaoming Ma, Qiang Zhang, Wei Zhang

**Affiliations:** 1School of Pharmacy, Changzhou University, 1 Gehu Road, Changzhou 213164, China; mxm.wuxi@cczu.edu.cn; 2School of Chemistry and Life Sciences, Suzhou University of Science and Technology, 99 Xuefu Road, Suzhou 215009, China; qzhang@mail.usts.edu.cn; 3Department of Chemistry and Center for Green Chemistry, University of Massachusetts Boston, 100 Morrissey Blvd, Boston, MA 02125, USA

**Keywords:** radical, difunctionalization, remote, 1,3-difunctionalization, 1,4-difunctionalization, 1,5-difunctionalization, 1,6-difunctionalization, 1,7-difunctionalization

## Abstract

Radical transformations are powerful in organic synthesis for the construction of molecular scaffolds and introduction of functional groups. In radical difunctionalization reactions, the radicals in the first functionalized intermediates can be relocated through resonance, hydrogen atom or group transfer, and ring opening. The resulting radical intermediates can undertake the following paths for the second functionalization: (1) couple with other radical groups, (2) oxidize to cations and then react with nucleophiles, (3) reduce to anions and then react with electrophiles, (4) couple with metal-complexes. The rearrangements of radicals provide the opportunity for the synthesis of 1,3-, 1,4-, 1,5-, 1,6-, and 1,7-difunctionalization products. Multiple ways to initiate the radical reaction coupling with intermediate radical rearrangements make the radical reactions good for difunctionalization at the remote positions. These reactions offer the advantages of synthetic efficiency, operation simplicity, and product diversity.

## 1. Introduction

Radical addition, coupling, rearrangement, and cleavage reactions are powerful in the synthesis of diverse molecular structures [1,2]. Difunctionalization reactions initiated with radical addition are attractive for both synthetic efficiency and product diversity considerations [3,4,5,6,7,8,9,10,11,12]. We have recently reported radical 1,2-difunctionalization (Figure 1I) [13] and addition-/cyclization-based difunctionalization reactions (Figure 1II) [14]. Compared to these two kinds of reactions, remote 1,3-, 1,4-, 1,5-, 1,6- and 1,7-difunctionalization reactions are new and under active investigation. Developments on this topic are highlighted in this paper. Most works have been published in the last five years.

In addition to traditional homolytic bond cleavage-based radical reactions induced by radical initiators or photolysis [15,16,17], other methods such as single electron transfer reagents [18,19,20,21,22], catalytic photoredox [23,24,25,26,27,28], and electrochemical reactions [29,30,31,32,33,34,35] have been developed and are gaining increasing popularity. For the remote 1,3-, 1,4-, 1,5-, 1,6- and 1,7-difunctionalization reactions presented in this paper, the initial addition of the radical **X** is followed by radical rearrangement through resonance, hydrogen atom transfer (HAT), group transfer, or opening of strained-rings to relocate the position of the radical. The resulting radical intermediates can couple with the radical **Y** to give the desired products. The radical intermediates can also be oxidized to cations, subsequently react with Y^−^ or reduced to anions, and then react with Y^+^ to give the products (Figure 1III).

## 2. 1,3-Difunctionalization Reactions

Substrates used for the 1,3-difunctionalization commonly have allyl or cyclopropyl moieties. Other special substrates, such as alkynyl diazo compounds and piperidines, can be used for the reactions (Figure 1). The general reaction pathways for 1,3-difunctionalization of allyl or cyclopropyl compounds are shown in Figure 2. The initial radical addition happens at the less hindered position of the substrate to form a stabilized radical intermediate after 1,2-group transfer or cyclopropyl ring opening, which then undergoes the second functionalization to give the product.

Studer and colleagues, in 2020, reported a method for the synthesis of 1,2,3-trisubstituted alkanes **1** using allylboronic esters as the substrates and acetylenic triflones as the reagent for 1,3-trifluoromethylacetylenic difunctionalization (Figure 3) [36]. In the reaction process, AIBN-initiated radical leads to the formation of CF_3_ radical from an alkynyl triflone which adds to the double bond of the allylboronic esters to form alkyl radicals **M-1** followed by 1,2-boron migration to give more stable radicals **M-2** which are trapped by acetylenic triflones to afford the products **1**. This methodology can be also applied to the 1,3-trifluoromethylazido difunctionalization using trifluoromethanesulfonyl azide to give product **1d**. This could also be considered as a trifunctionalization reaction since it involves the migration of the boronic ester group.

*β*-Alkyl nitroalkenes are good substrates for 1,3-difunctionalization. Shi and Chen group employed them in the reactions with TEMPO for the synthesis of ketones **2** bearing the vinylic alkoxyamine group (Figure 4) [37]. In the reaction process, *β*-alkyl nitroalkenes are isomerized to allylic nitro compounds **M-3** in the presence of a Lewis base, which are oxidized by TBHP to form radical intermediates **M-4**. The coupling of **M-4** with TEMPO gives **M-5** followed by the addition of *t*BuOO^−^ to form functionalized ketones **2** after NO_2_ elimination.

Opening of the cyclopropane ring is a good strategy for 1,3-difunctionalization. In 2021, Lei et al., reported an electrochemical reaction of arylcyclopropanes for the synthesis of 1,3-difluorinated compounds **3** using Et_3_N.3HF as a reagent for difluorination (Figure 5) [38]. Arylcyclopropane radical cations **M-6** generated from the anode oxidation of arylcyclopropanes react with a nucleophile to form radicals **M-7** which are oxidized to carboniums **M-8** and then react with the second nucleophiles to afford 1,3-diflourinated products **3**. If alcohols or ethers are used as the nucleophiles, the reaction can produce 1,3-fluoroalkoxylated products **4a**–**f** and 1,3-dialkoxylated products **4g**–**h**, respectively.

Other than the allylic and cyclopropyl compounds presented above, alkynyl diazo compounds can be used for the synthesis of functionalized allenes. Zhu et al., in 2022, reported a visible-light-promoted radical reaction of alkynyl diazo compounds with RSO_2_X for the synthesis of tetrasubstituted allenes **5** in high yields (Figure 6) [39]. A proposed mechanism suggests that the sulfonyl radical adds to the *α*-position of the alkynyl diazo compounds to form allenyl radicals **M-9** after elimination of N_2_. Coupling of allenyl radicals with X radicals from RSO_2_X gives 1,3-difunctionalized allenes **5** and release the sulfonyl radicals simultaneously. The resulting tetrasubstituted allenes are good substrates that can be used for cascade Michael addition/cyclization reactions for making cyclobutanone derivatives.

A unique α,γ-difunctionalization reaction of *N*-aryl piperidines for making bridged products was reported by Zhou et al., in 2022 (Figure 7) [40]. In the presence of 3DPAFIPN and under blue light photocatalysis, radical **M-10** generated from nitrobenzene reacts with **M-11** to form iminium intermediate **M-12** which is then oxidized to **M-13**. Base-promoted formation of *N*-radical **M-14** from **M-13** undergoes 1,5-HAT to form radical **M-15** which is then coupled with the *N*-radical intramolecularly to give final product **6a**.

The Xia and Guo group, in 2022, reported 1,3-difunctionalization of a special kind of substrates, alkyl *N*-hydroxyphthalimide esters, in the synthesis of *γ*-cyano alkenes **7** or *γ*,*δ*-unsaturated ketones **8** (Figure 8) [41]. Under visible-light-induced photochemical conditions, alkyl radicals **M-16** resulting from alkyl *N*-hydroxyphthalimide esters add to alkenes to form radicals **M-17** which then undergo 1,5-HAT to form radicals **M-18**. If TMSCN is used for the reaction, radicals **M-18** are converted to complex **M-19** followed by reductive elimination to give *γ*-cyano alkenes **7**. If DMSO instead of TMSCN is used for the reaction, radicals **M-18** are oxidized to cations **M-20** and then react with DMSO to form *γ*,*δ*-unsaturated ketones **8**. It is worth noting that path A (with TMSCN) requires Cu catalyst, while path B (with DMSO) does not need Cu catalyst.

In 2020, Chu and colleagues reported a unique cyclic-oxalate-based reaction involving decarboxylative vinylation/1,5-HAT/aryl cross-coupling for the synthesis of α,γ-difunctionalized cyclohexanes **9** under photoredox and Ni dual catalysis (Figure 9) [42]. In the reaction process, the Ir-catalyzed photoredox reaction of cyclic oxalates promotes the decarboxylation to form radicals which add to alkynes to form vinyl radicals **M-21**. The 1,5-HAT of **M-21** followed by coupling with LNi^0^ and then with ArBr afford complexes **M-22**. Reductive elimination of the Ni-cat produces product **9**.

## 3. 1,4-Difunctionalization Reactions

1,4-Difunctionalizations are more popular than 1,3-difunctionalizations. The common substrates for 1,4-difuntionalizations include 1,3-dienes, 1,3-enynes, pent-1-ynes, and isoquinolines (Figure 2). The general reaction pathways for the reaction of 1,3-dienes, 1,3-enynes, and pent-1-ynes are shown in Figure 10. After addition of **X**, the position of intermediate radicals is relocated through resonance or 1,5-HAT to the 4-position for second functionalization with **Y** to give the 1,4-difunctionalization products.

Song and Li’s group, in 2020, reported the difunctionalization of 1,3-dienes with alkyl radicals and heterocyclics nucleophiles. The reaction of aromatic 1,3-dienes, *α*-carbonyl alkyl bromides and *N*-heterocycles in the presence of InBr_3_ and Ag_2_CO_3_ afforded substituted *N*-heterocycles **10** in moderate to good yields (Figure 11) [43]. However, the aliphatic 1,3-diene was inactive. A proposed mechanism indicated that In-coordinated alkyl radical **M-23**, generated from (CH_3_)_2_BrCO_2_Et via SET of Ag_2_CO_3_ and the In catalyst, adds to the terminal carbon of the 1,3-diene to form ŋ^3^-allyl-In radical complex **M-24** followed by single-electron oxidation to form cation **M-25** which then reacts with heterocyclic nucleophile to afford product **10a** as the major product.

A visible-light-mediated and Pd-catalyzed reaction of 1,3-dienes was reported by Glorius et al., in 2020. Under the radiation of blue LEDs and in the presence of Pd(PPh_3_)_4_, BINAP and KOAc in DMA, a three-component reaction of 1,3-dienes, alkyl bromides and nitrogen-, oxygen-, sulfur-, or carbon-based nucleophiles afforded products **11** in good to excellent yields (Figure 12) [44]. The reaction mechanism suggests that hybrid alkyl Pd^I^ radical **M-26**, generated from *tert*-butyl bromide by photoinduced Pd catalysis, adds to the C=C bond of butadiene to form allyl Pd^I^-radical complex **M-27** and then Pd^II^-complex **M-28** after SET. The reaction of **M-28** with a nucleophile followed by reductive elimination of the Pd-cat gives product **11a**.

In 2021, Wang et al., reported a Ni-catalyzed three-component reaction of 1,3-butadiene with ethyl 2-bromo-2,2-difluoroacetate and arylboronic acids for the synthesis of 1,4-difluoroalkylarylated products **12** in good to excellent yields. However, *ortho*-substituted phenylboronic acids and cyclohexylboronic acid were found inert (Figure 13) [45]. A proposed reaction pathway indicates that the CF_2_CO_2_Et radical derived from BrCF_2_CO_2_Et adds to the C=C bond of 1,3-butadiene to form allyl radical **M-29** which reacts with ArNi^II^LBr complex to form Ni^III^ intermediate **M-30** followed by reductive elimination to give product **12a**.

In 2022, Yang et al., reported a visible-light-induced and Pd-catalyzed reaction of 1,3-dienes with bromodifluoroacetamides and sulfinates or amines for the synthesis of difluorofunctionalized alkenes **13** in moderate to good yields (Figure 14) [46]. The reaction mechanism suggests that hybrid alkyl Pd^I^ radical **M-31**, generated from BrCF_2_CONHPh via a SET process by photo-induced Pd catalysis, adds to the terminal position of 1,3-butadiene to form hybrid allyl Pd^I^-radical **M-32** followed by SET for a Pd^II^- complex which then undergoes nucleophilic addition and reductive elimination of the Pd catalyst to give product **13a**.

Other than 1,4-difunction of 1,3-dienes for making substituted but-2-enes presented above, 1,3-enynes are important substrates for 1,4-difuncntion for the synthesis of substituted allenes. The key reaction process involves the resonance of propargyl radicals to allenyl radicals for the second functionalization. There are many examples reported in the literature including asymmetric synthesis under photoredox catalysts, transition metal-catalysts, and organocatalysts.

In 2009, Kambe and colleagues developed a transition metal-catalyzed reaction of 1,3-enynes, alkyl halides, and organozinc reagents for regioselective synthesis of 1,4-difunctionalized allene product **14** in moderate to good yields (Figure 15) [47]. A proposed mechanism for 1,4-difunctionalization of 1,3-enynes indicated that the Ni(dppb) species generated from Ni(acac)_2_ and organozinc reagents reacts with R_2_Zn to afford Ni-complex **M-33**. The alkyl radical generated from the Ni-complex **M-33** adds to the 1,3-enynes at the terminal position of the olefin followed by resonance to form allenyl radical intermediate **M-34**. The allenyl radical is trapped by (dppb)Ni^I^-R complex to give Ni-complex intermediate **M-35** which then undergoes reductive elimination to afford allene product **14**.

In 2019, Wang et al., reported a method for the synthesis of 1,4-fluoroalkylated allenes **15** via a Ni-catalyzed reaction of 1,3-enynes under mild conditions (Figure 16) [48]. In the reaction process, the arylated Ni^I^ species **M-36** generated through the transmetallation of the Ni^I^ catalyst with aryl boronic acid reduces the fluoroalkyl bromide to afford fluoroalkyl radical **M-37** and Ni^II^ complex **M-38**. The capture of fluoroalkyl radical **M-37** by a 1,3-enyne followed by 1,3-radical shift generates the key intermediate allenyl radical **M-39**, which then coordinates with Ni^II^ complex **M-38** to afford oxidized Ni^III^ complex **M-40**. At the last step, reductive elimination of Ni^III^ complex **6** gives the tetrasubstituted allene product **15**.

In 2019, Ma et al., reported a Cu-catalyzed atom transfer radical addition of aryl sulfonyl iodides to 1,3-enynes for the synthesis of allenyl iodides **16** under mild conditions (Figure 17) [49]. A suggested reaction mechanism indicates that the aryl sulfonyl radicals (ArSO_2_·) generated from aryl sulfonyl iodides adds to the alkene moiety of 1,3-enynes to afford allenyl radical **M-41**. Trapping of the radicals **M-41** with LCuI_2_ produces allenyl Cu^III^ diiodide species **M-42** which lead to the formation of allenyl iodides **16** after reductive elimination of the catalyst.

Recently, Lv and colleagues developed a Cu-catalyzed 1,4-sulfonylcyanation reaction of 1,3-enynes with alkyl or aryl sulfonyl chlorides and TMSCN (Figure 18) [50]. Under the catalysis of Cu(CH_3_CN)_4_PF_6_, sulfonyl-containing allenic nitriles **17** were obtained in good yields and high regioselectivity. A reaction mechanism suggests that the sulfonyl radicals generated from sulfonyl chlorides and LCu^I^ species adds to the alkene moiety of 1,3-enynes to afford allenyl radicals **M-43** followed by the coordination with LCu^II^Cl and ligand exchange with TMSCN to give cyano-Cu^III^ species **M-44**. Sulfonyl-containing allenyl nitrile products **17** are obtained after the reductive elimination of the Cu-catalyst. Lv et al., also developed a 1,4-sulfonyliodination reaction of 1,3-enynes to synthesize a tetrasubstituted allenyl iodides **18** under metal-free conditions (Figure 19) [51]. The reaction of 1,3-enynes with sulfonyl hydrazides and I_2_ in the presence of *tert*-butyl hydroperoxide (TBHP) at room temperature gave the allenyl iodide products in satisfactory yields with excellent regioselectivity and good functional group tolerance. In 2022, Li and Wang’s group disclosed a visible-light-induced and Ni-catalyzed 1,4-arylsulfonation of 2-methyl-1,3-enynes to synthesize compounds **19** (Figure 20) [52].

A number of Cu-catalyzed and Togni’s reagent-based trifluomethylation reactions have been reported. In 2018, Liu et al., reported a tunable 1,2- and 1,4-addition of 1,3-enynes for the synthesis of CF_3_-containing tri- and tetrasubstituted allenyl nitriles (Figure 21) [53]. The Cu-catalyzed reaction of 1,3-enynes with Togni’s reagent and TMSCN in the presence of Cu(CH_3_CN)_4_PF_6_ under nitrogen atmosphere gave 1,2-/1,4-addition allenyl nitriles (**20**/**21**) in moderate to excellent yield. The regioselectivity could be controlled by using different ligands. The reactions using phenanthroline-type ligand **L1** primary gave 1,4-addition allenyl nitriles product **20** through an allenyl-Cu^III^ species **Int-I**. The reactions using bisoxazoline ligands **L2** in the presence of Et_3_N produced 1,2-propargylic cyanation products **21** via the **Int-II** complex intermediates. It is worth noting that, the reactions of 1,3-enynes with R^2^ at C2 position only afforded 1,4-addition product **20c** due to the steric hindrance at C2 position which prevents the interaction of the tertiary propargyl radical with the reactive Cu^II^ cyanide complex. In 2020, the Li group reported a Cu-catalyzed reaction of 1,3-enynes with Togni II reagent and (bpy)Zn(CF_3_)_2_ for the synthesis of 1,4-bis(trifluoromethylated) allenes **22** (Figure 22) [54].

Yang and colleagues, in 2021, extended the reaction scope for the synthesis CF_3_-containing tetrasubstituted allenes. They reported a Cu-catalyzed 1,4-difunctionalization reaction of 1,3-enynes with Togni II reagent and a nucleophilic halide reagent (SOX_2_) (Figure 23) [55]. In the reaction process, a CF_3_ radical generated from Togni II adds to the 1,3-enynes to afford allenyl radical intermediates **M-45** followed by the combination with Cu^II^ and SOCl_2_ to give a CF_3_-allenyl-Cu^III^Cl_2_ species **M-46**. Reductive elimination of the Cu-cat gives 1,4-halotrifluormethylation product **23**.

The Yang and Cao group, in 2023, reported a Cu-catalyzed ATRA reaction of 1,3-enynes with Togni II reagent for making trifluoromethylbenzoxylated allenes **24** (Figure 24) [56]. The Togni II reagent plays triple roles in the reaction process, including the source of CF_3_ radical, the nucleophile for the second functionalization, and an oxidant for Cu catalysis. It is worth noting that 1,3-enynes bearing the fully substituted alkene moiety were employed to disfavor the radical addition to the alkene moiety at the initiate step. Thus, in this reaction system, CF_3_ radical attacks the alkyne position of 1,3-enynes to generate trifluoromethyl-substituted allenyl radical **M-47** which are oxidized to cations **M-48** followed by nucleophilic addition to form product **24**. The products can be readily converted to corresponding allenols **25**.

In 2021, Ma et al., introduced a Cu-catalyzed 1,4-addition of 1,3-enynes with cyclobutanone oxime esters and TMSCN to give allene products **26** in moderate to good yields (Figure 25) [57]. A reaction mechanism suggests that the Cu^I^ species reacts with cyclobutanone oxime ester to give the cyanoalkyl radical **M-49** and Cu^II^ species **M-50** via a SET process. Radical **M-49** adds to the C=C bond of 1,3-enynes to afford allenyl radicals **M-51**. In another path, intermediate **M-52**, which is generated by ligand exchange of **M-50** with TMSCN, couples with allenyl radicals **M-51** to produce allenyl Cu^III^ complex **M-53**. Subsequential reductive elimination of LCu^III^Br affords 1,4-carbocycanated allene products **26**. If TMSCF_3_ is used to replace TMSCN as a nucleophile, the reactions give 1,4-carbotrifluoromethylated allene products.

Wu and colleagues, in 2021, disclosed a Cu-catalyzed reaction of 1,3-enynes to give cyanoalkylsulfonylated allenyl selenides products **27** (Figure 26) [58]. The reaction of 1,3-enynes, diselenides, DABCO·(SO_2_)_2_ and cycloketone oxime esters under the catalysis of CuOAc without ligand gave products **27** in good yields. A reaction mechanism suggests that the iminyl radicals generated from cycloketone oxime esters undergoes *β*-C–C bond cleavage to give cyanoalkyl radical **M-54** which then is captured by SO_2_ from DABCO·(SO_2_)_2_ to generate cyanoalkylsulfonyl radical **M-55**. The addition of radical **M-55** to 1,3-enynes at the terminal C=C bond carbon affords propargyl radical which is converted to allenyl radical **M-56** through resonance. The coordination of **M-56** with Cu^I^ specie gives Cu^II^ complex **M-57** followed by the interaction with diphenyl diselenide to afford Cu^III^ complex **M-58** and a phenyl seleno radical. Subsequent reductive elimination of Cu^III^ affords product **27a**.

The vinyl enynes are good substrates for 1,4-difunctionalization. In 2021, Wu et al., reported a visible-light-induced 1,4-hydroxysulfonylation of vinyl enynes for the synthesis of sulfonyl allenic alcohols [59]. The reaction of diarylterminated enynes and aryl or alkyl sulfonyl chlorides in the presence of *fac*-Ir(ppy)_3_ and K_3_PO_4_ under the radiation of blue LEDs afforded 1,4-hydroxysulfonyl allenes **28** in good to excellent yields (Figure 27). However, the action with trifluoromethanesulfonyl chloride for **28e** was ineffective. A reaction mechanism suggests that the hydroxyl radical generated from water via the HAT with chloride radical adds to the C=C bond of diarylterminated enyne to form the propargyl radical followed by tautomerization to the allenic radical **M-59** which couples with the sulfonyl radical to give product **28a**.

Wu et al., in 2022, reported a metal-free radical difunctionalization reaction of vinyl enynes with NBS to afford diverse 4-bromo-allenic alcohols **29** in good yields (Figure 28) [60]. In the reaction process, hydroxyl radical derived from H_2_O adds to the C=C bond and then traps bromo radicals generated from NBS to give products **29**.

*N*-Hydroxyphthalimide (NHP) esters are good precursors for generating alkyl radicals. In 2021, Lu et al., reported a photoredox and Cu-catalysis reaction for 1,4-carbocyanation of 1,3-enynes (Figure 29) [61]. The reaction of 1,3-enynes with *N*-hydroxyphthalimide (NHP) esters and TMSCN under Cu/photoredox dual catalysis gave tetrasubstituted allenes **30** in good yields and excellent functional group tolerance. A reaction mechanism suggests that the excited state Ir^III*^ generated from photocatalyst Ir(ppy)_3_ reacts with the NHP esters through a SET process to give ester radical anions **M-60** which undergo decarboxylation to form alkyl radicals R^3^·. The subsequent alkyl radical addition to C=C double bond of 1,3-enynes affords allenyl radical **M-61** which coordinates with TMSCN to create cynanocopper^III^ species **M-62**. At the last step, reductive elimination of the Cu-catalyst affords tetrasubstituted allenic nitriles **30**.

There are several reports on dual Ni/photoredox catalyzed reactions for 1,4-difunctionalizations. Lu et al., reported 1,4-sulfonylarylation of 1,3-enynes for the synthesis of allenes **31** (Figure 30) [62]. In the presence of NiCl_2_·glyme, diOMebpy ligand, organic photosensitizer (1,2,3,5-tetrakis(carbazol-9-yl)-4,6-dicyanobenzene) (4CzIPN), the reaction of 1,3-enynes with sodium sulfinates as radical precursors and aryl halides as coupling partners afforded sulfone-containing allenes **31** in fair to good yields. A reaction mechanism suggests that the excitation of photocatalyst (PC) 4CzIPN leads to the activated **M-63** which then reacts with R^3^SO_3_Na to form sulfonyl radical R^3^SO_2_· along with the reduced state PC **M-64**. The addition of sulfonyl radicals to the alkene moiety of 1,3-enynes affords allenyl radicals **M-65** which then convert to allenyl Ni^I^ species **M-66** after interception with LNi^0^. Oxidative addition of aryl halides to **M-66** creating the allenyl Ni^III^ intermediates **M-67** which undergo reductive elimination to give sulfone-containing allenes **31**. The Li and Wang groups also reported the 1,4-sulfonylarylation of 1,3-enynes [63,64].

In 2022, the Kong and Wang groups introduced a photoredox reaction for dicarbonation of trifluoromethylated 1,3-enynes (Figure 31) [65]. In the presence of TBADT and Ni(dibbpy)Br_2_ catalysts and under near-ultraviolet light irradiation, the reaction of 1,3-enynes with alkanes and alkyl halides afforded tetrasubstituted CF_3_-allenes **32**. A reaction mechanism suggests that the cyclohexyl radical generated by the photoredox catalysis adds to the alkene moiety of 1,3-enyne to give allenyl radical **M-68** which reacts with **M-69** to form Ni-complex **M-70** followed by a Ni shift to give more stable allenyl-Ni^I^ complex **M-71**. The subsequent oxidative addition of ethyl 4-bromobenzoate to **M-71** gives allenyl-Ni^III^ intermediate **M-72** which gives product **32a** after reductive elimination of LNi^III^Br.

In 2021, Du et al., introduced an *N*-heterocyclic carbene (NHC) organocatalyzed reaction for 1,4-alkylcarbonylation of 1,3-enynes **33** (Figure 32) [66]. The reaction of 1,3-enynes with alkyl radical precursors and aldehydes under NHC organocatalysis gave allenyl ketone products **33** in moderate to excellent yields. A reaction mechanism suggests that the reaction of an aldehyde and NHC **M-73** under a basic condition gives Breslow intermediates **M-74** which interact with CF_3_I to afford a CF_3_ radical and NHC-bound ketyl radicals **M-75**. The addition of CF_3_ radical to the C=C double bond of 1,3-enynes gives allenyl radicals **M-76** which are coupled with ketyl radicals **M-75** to give NHC-allenyl intermediates **M-77**. The final allenyl ketone products **33** are generated from **M-77** after elimination of NHC. Recently, Du et al., applied a similar reaction for making *gem*-difluorovinylcarbonylated allenes **34** (Figure 33) [67]. The Huang and Yang groups also reported this kind of reactions for the synthesis of 1,4-alkylcarbonylated allenes **35** (Figure 34) [68,69]. In 2022, the Zhang and Zheng’s group reported a reaction which combined the NHC and photoredox catalysis for 1,4-sulfonylacylation of 1,3-enynes. The reaction of 1,3-enynes, aroyl fluorides, and sodium sulfinates under blue light irradiation affords sulfone-containing allenyl ketones **36** in moderate to good yields (Figure 35) [70].

In 2019, Bao et al., reported a Cu-catalyzed 1,4-alkylarylation of 1,3-enynes using diacyl peroxide as radical precursors and aryl boronic acids as nucleophiles to afford tetrasubstituted allenes **37** in moderate to good yields (Figure 36) [71]. In the reaction process, LCu^II^-Ar complexes **M-78** and alkyl radicals are resulting from (RCO_2_)_2_. The alkyl radicals add to 1,3-enynes to form allenyl radicals **M-79** which then react with **M-78** in two possible ways. In path a, the radicals **M-79** couple with **M-78** to form tetrasubstituted allene products **37** and regenerate the LCu^I^ catalyst. While in path b, the coordination of **M-79** with **M-78** afford Cu^III^ species **M-80** which then give product **37** after reductive elimination of LCu^I^ catalyst.

Bao et al., in 2019, reported another Cu-catalyzed reaction for 1,4-alkylcyanation, 1,4-fluoroalkylcyanation and 1,4-sulfimidocyanation of 1,3-enynes (Figure 37) [72]. The reaction of 1,3-enynes with TMSCN and various radical precursor reagents (alkyl diacyl peroxides, fluoroalkylated iodides and *N*-fluorobenzenesulfonimide) afforded corresponding allenyl nitriles compounds **38** as racemic compounds in moderate to good yield and high regioselectivity. The proposed reaction mechanism is different from the Cu^III^ mechanism in the cyanation reactions reported previously. In this case, the alkyl radical generated from diacyl peroxide add to 1,3-enynes to form allenyl radicals **M-81** which then couple with isocyanocopper species **M-82** to form complexes **M-83**. Reductive elimination of LCu^II^ catalyst from **M-83** affords substituted allenyl nitriles **38**. In 2020, Bao et al., employed the use of a chiral ligand for asymmetric 1,4-difunctionalization of 1,3-enynes in the synthesis of **39** (Figure 38) [73].

Organocatalysts can be used for asymmetric 1,4-carboalkynylation of 1,3-enyne. Liu et al., in 2019, reported a Cu and cinchona alkaloid-derived catalytic system for the reaction of 1,3-enynes with alkyl halides and alkynes to give 1,4-carboalkynylation products **40** in moderate to good yields with the high *ee* ratio (Figure 39) [74]. A reaction mechanism suggests that complex **M-84**, generated from the reaction of Cu^I^X, ligand and alkynes under basic conditions, reacts with alkyl halides to form Cu^II^ species **M-85** and R^3^ alkyl radicals. The alkyl radicals add to the 1,3-enynes to form allenyl radicals **M-86** then couple with **M-84** to afford chiral tetrasubstituted allenes **40**. In this reaction, the chiral cinchona alkaloid-derived N,N,P-complex is the key for the enantiocontrol during the reaction with highly reactive allenyl radical **M-86**.

Wang et al., in 2022, reported an asymmetric 1,4-difunctionalization of 1,3-enynes using dual photoredox and Cr catalysts. The reaction of 1,3-enynes with aldehydes and DHP esters in the presence of CrCl_2_, 4CzIPN and chiral ligand under the radiation of blue LEDs afforded chiral allenols **41** in moderate to good yields with high enantioselectivities (Figure 40) [75]. The reaction mechanism suggests that isopropyl radical generated from DHP ester assisted adds to 1,3-enyne to provide propargyl radical **M-87** followed by trapping with Cr^II^L to form the propargyl chromium **M-88** and then chiral intermediate **M-89** after nucleophilic addition to benzaldehyde. A six-member cyclic transition state controls the enantioselectivity for the Nozaki–Hiyama allenylation [76]. The final product **41a** is then obtained after the protonation of **M-89**.

Highly strained alkylidenecyclopropanes (ACPs) are useful structure moieties in organic synthesis. Sequential 6π-electrocyclization and vinylcyclopropane rearrangement of allene-type ACP intermediates can afford more stable aromatization heterocyclic products. In 2020, Shi and coworkers reported a Cu-catalyzed 1,4-difunctionalization reaction of 1,3-enyne-ACPs with Togni I reagent and TMSCN to afford 3-trifluoroethylcyclopenta[*b*]naphthalene-4-carbonitrile derivatives **42** in moderate to good yields (Figure 41) [77]. The proposed mechanism indicated that the CF_3_ radical generated from Togni I reagent added to the 1,3-enyne-ACPs to form allenyl radicals **M-91** after tautomerization. Allenyl radicals **M-91** are captured by the LCu^II^-CN complex followed by the reductive elimination to produce allene-ACP products **M-92**. The 6π-electrocyclizaton of **M-92** gives vinylcyclopropane intermediates **M-93** which undergoes CN-catalyzed cyclopropane ring opening and *S_N_2* cyclization to give cyclopenta[*b*]naphthalene products **42**. In addition, the interaction of vinylcyclopropane intermediate **M-93** with cyano anions, followed by the further *S_N_2* reaction also can give the final product **42**.

Other than the popular conjugated 1,3-dienes and 1,3-enynes presented above, special aromatic substrates can also be developed for 1,4-functionalization reactions. Yan et al., in 2018, introduced a Cu-catalyzed reaction to 1,4-difunctionalize the isoquinolinium salts with ethers and halogen anions. The reaction of isoquinolinium salts and esters in the presence of Cu(acac)_2_ and TBHP afforded substituted azaarenes **43** in moderate to good yields [78]. However, the dioxane and diethyl ether were found to be less reactive (Figure 42). The reaction mechanism suggests that the THF radical, generated from the oxidation of THF and *tert*-butyl hydroperoxide (TBHP) through a Cu-catalyzed process, adds to the C-1 position of 2-benzylisoquinolin-2-ium bromide to form the radial cation **M-94** followed by radical resonance to form **M-95** which then undergoes a Cu-catalyzed bromine radical coupling and deprotonates to give product **43**.

Fullerene is another aromatic substrate which has been used for 1,4-difunctionalization reaction. Jin et al., in 2015, reported a reaction of C_60_ with benzyl bromides under the Ni-catalysis to afford 1,4-dibenzyl fullerene compounds **44** in good yields (Figure 43) [79]. Using a cosolvent for the NiCl_2_dppe catalysis is essential for the success of this reaction. As shown in the proposed mechanism, Ni^0^L species generated from the reduction of Ni^II^L with Mn reacts with benzyl bromide to form benzyl radicals which add to C_60_ to afford the fullerene radical **M-96** followed by the subsequent coupling with another benzyl radical species to give 1,4-dibenzyl fullerenes **44**.

As shown in Figure 2, pent-1-yne derivatives are also good substrates for radical 1,4-difunctionalizations through a critical 1,5-HAT process. Zhu et al., in 2020, introduced a photoredox reaction of heteroalkynes using oxyfluoroalkylation as radical source and using DMSO or H_2_O as nucleophiles to afford oxyfluoroalkylated (*Z*)-alkenes **45**/**46** in moderate to good yields (Figure 44) [80]. The CF_3_ radical generated from the Umemoto’s reagent adds to the *β*-carbon of heteroalkynes to form the vinyl radical **M-97** which then undergoes 1,5-HAT to give alkyl radicals **M-98**. Oxidation of **M-98** to alkyl cations **M-99** followed by nucleophilic attack with DMSO or H_2_O leads to the formation of (*Z*)-alkenol **45c** and **46a**, respectively.

In 2020, Zhu et al., reported an Ag-mediated fluoro-fluoroalkylation reaction of alkynes [81]. The reaction of alkynes with fluoroalkyltrimethylsilanes (TMSRf) and Selectfluor in the presence of AgNO_3_, PhI(OCOCF_3_)_2_ (PIFA) and CsF afforded *γ*-fluorinated fluoroalkyated (*Z*)-alkenes **47** in good yields. However, the reaction of thioalkyne was ineffective (Figure 45). A proposed reaction pathway indicates that the CF_2_CO_2_Et radical derived from TMSCF_2_CO_2_Et adds to an internal alkyne to form the vinyl radical **M-100** and then 1,5-HAT to form **M-101** followed by Ag-assisted fluorination with Selectfluor reagent to give the product **47a**. This reaction can also be conducted under photoredox conditions.

In 2020, Zhu et al., reported an azobisisobutyronitrile (AIBN)-induced trifluoromethyl-alkynylation reaction of thioalkynes to make trifluoromethylated (*Z*)-enynes **48** in moderate to high yields with excellent regio- and stereoselectivity (Figure 46) [82]. Moreover, the base treatment of (*Z*)-enyne **48a** provided trifluoromethyl allene **49**. The desilylation of **48b** with TBAF followed by a Cu-catalyzed click reaction afforded the trifluoromethyl triazole product **50**. A reaction mechanism indicates that the CF_3_ radical, generated from the reaction of PhC≡CSO_2_CF_3_ and AIBN, adds to the *β*-carbon of thioalkyne to form a vinyl radial followed by 1,5-HAT to produce the alkyl radical **M-102** which then adds to the electrophilic carbon triple bonds of PhC≡CSO_2_CF_3_ to yield a new vinyl radical **M-103** followed by the *β*-elimination to give product **48a**.

In 2014, Taniguchi et al., reported a unique 1,4-dihydroxylation of terminal and internal alkenes for the synthesis of 1,4-diols **51**/**52** via 1,5-HAT of oxygen radicals (Figure 47) [83]. A proposed mechanism indicates that the interaction of Fe phthalocyanine complex with NaBH_4_ and O_2_ gives a putative Fe^III^ hydride complex which adds to alkenes to afford intermediates **M-104**. The reaction of **M-104** with O_2_ produces tertiary-carbon-centered radicals which are captured by the Fe-complex to afford the iron peroxide complex **M-105**. The formation of alkoxy radical via the cleavage of the O–O bond of **M-105** and 1,5-HAT of the O-radicals give alkyl radicals **M-106** which lead to the formation of 1,4-diols **51** after reduction of the radicals and capture of the second O_2_.

## 4. 1,5-Difunctionalization Reactions

All the radical 1,5-difunctionalization reactions summarized in this paper have been published within the last four years and the numbers are limited. These reactions require special substrates that contain vinyl cyclopropane or 5-membered rings with two heteroatoms. The ring-opening to relocate the radicals is the key reaction process which allows the second functionalization at the 5-position (Figure 3). A representative pathway for the reaction of vinyl cyclopropanes is shown in Figure 48. In the reaction process, the addition of initial **X** radical generates cyclopropylmethyl radicals which readily open to form new radicals for the second functionalization with **Y** to give the products.

An enantioselective 1,5-cyanotrifluoromethylation of vinylcyclopropanes (VCPs) through a Cu-catalyzed radical reaction was reported by Wang et al., in 2019 [84]. The reaction of VCPs, Togni’s reagent I and TMSCN in the presence of Cu(acac)_2_ and chiral oxazoline ligand afforded CF_3_-containing alkenylnitriles **53** in good yields and *ee* ratio (Figure 49). The reaction mechanism shows that CF_3_ radical derived from the Togni I reagent adds to the C=C bond of vinylcyclopropane to form alkyl radical **M-107** and then benzylic radical **M-108** after *β*-fragmentation of the cyclopropane ring. The enantioselective reaction of **M-108** with chiral LCu^II^(CN)_2_ affords product **53a** after reductive elimination of the catalyst.

The Cahard group, in 2021, reported a vinylcyclopropane (VCP)-based 1,5-chloropentafluorosulfanylation for the synthesis of allylic pentafluorosulfanyl derivatives. The reaction of VCPs and SF_5_Cl in alkanes in the presence of Et_3_B and O_2_ gave products **54** in high yields (Figure 50) [85]. The reaction mechanism suggests that SF_5_ radical, generated from the reaction of SF_5_Cl with Et_3_B and O_2_, adds to the C=C bond of VCPs followed by cyclopropane ring-opening and coupling with chlorine radical of SF_5_Cl to provide 1,5-chloropentafluorosulfanylation product **54a**.

The Yan group, in 2019, reported a benzothiazolim-bromide-based difunctionalization reaction. The Cu-catalyzed reaction of benzothiazolim bromides with benzodioxole afforded 2,5-difunctionalized benzothiazolims **55** in moderate to good yields (Figure 51) [86]. This reaction initiates with the addition of benzodioxole radical to benzothiazolims at the 2-position followed by resonance relocation of the radical from N atom to 6-position and then oxidative coupling with [Cu]-Br to give products **55**.

The Feng group, in 2022, reported the use of 3-alkyl-4-isoxazolines as substrates for a photoredox 1,5-difunctionalization reaction to make *α*-sulfonyl-*β*-amino ketone and *α*-polyfluoroalkyl-*β*-amino ketone compounds **56** in good to excellent yields (Figure 52) [87]. The reaction was applied for the preparation of enantiopure *α*-polyfluoroalkyl-*β*-amino ketone **57** as well as Fe-catalyzed trifluoromethylation-azidation reaction for making product **58**. A reaction mechanism suggests that PhSO_2_ radical adds to the 4-position of 4-isoxazoline for radical-addition-induced *β*-fragmentation (RAIF) to cleave the N–O bond followed by 1,5-HAT and trifluoromethylthiolation to give product **56a**.

## 5. 1,6- and 1,7-Difunctionalization Reactions

Radical 1,6- and 1,7-difunctionalization reactions require special alkene substrates which can undergo 1,5- or 1,6-HAT reactions (Figure 4). Since a couple of recent reviews covered the progress on this topic [88,89], only selective examples and most recent examples are presented herein.

As presented in this paper above (Figure 44, Figure 45 and Figure 46), 1,5-HAT is also involved in the 1,4-difunctionalization reactions. In the case of 1,4-difunctionalization, the initial radical addition happens at the alkyne carbon to form R^1^Z-stabilized vinyl radicals which undergo 1,5-HAT to shift the radical to carbon-4 of the initial addition (Figure 53). Meanwhile in the 1,6-difunctionalization, the initial addition happens at the terminal carbon of alkenes to give alkyl radicals which undergo 1,5-HAT to shift the radical to carbon-6 of the initial addition.

In 2014, the Liu group reported an asymmetric 1,6-alkoxytrifluoromethylation reaction of alkenes under the Cu and chiral phosphoric acid (CPA) co-catalysis [90]. The reaction of alkenes, Togni’s reagent I and alcohols gave the chiral CF_3_-containing *N*,*O*-aminals **59** in good yields with excellent enantioselectivities (Figure 54). A reaction mechanism suggests that the CF_3_ radical derived from the Togni’s reagent I adds to the terminal carbon of alkenes. The resulting radical **M-109** undergoes 1,5-HAT followed by oxidation to give imine compound **M-110**. The CPA-catalyzed nucleophilic attack of MeOH on imine **M-110** affords the chiral product **59a**. It is worth noting that when indoles were used as the nucleophiles instead of alcohols under the Cu-CPA catalytic system, a series of chiral trifluoromethylated indole derivatives could be obtained [91]. Similar Cu-catalyzed reactions for racemic products [92] and metal catalyst-free 1,6-difunctionalization of alkenes [93] were also developed by the same group.

Liu and colleagues, in 2015, reported a Cu-catalyzed 1,6-difunctionalization reaction of alkenes to introduce azido and CF_3_ groups [94]. The reaction of alkenyl ketones, TMSN_3_ and Togni’s reagent II in the presence of CuI afforded 1,6-azidotrifluoromethylation products **60** in good to excellent yields. The CF_3_ radical derived from Togni II adds to the terminal carbon of alkene followed by 1,5-HAT to give radical **M-111** which is then oxidized by Cu^II^ to provide cation intermediate **M-112**. Nucleophilic reaction of **M-112** with TMSN_3_ affords product **60a** (Figure 55).

In 2015, the Liu and Tan group reported a 1,2-bis(diphenylphosphino)benzene (dppBz)-promoted reaction of alkenes with Togni’s reagent II to give 1,7-bistrifluoromethylated enamides **61** in excellent yields with good regio-, chemo-, and stereoselectivities (Figure 56) [95]. In the reaction process, the CF_3_ radical derived from the Togni’s reagent II adds to alkenes followed by 1,5-HAT to afford a more stabilized *α*-amido radicals **M-113**. After single-electron oxidation of **M-113** with Togni’s reagent II and deprotonation afford enamides **M-114** which react with second CF_3_ radical followed by single-electron oxidation to radical cations and deprotonation to furnish 1,7-bistrifluoromethylated enamides **61**.

In 2018, Nevado and colleagues described a redox neutral remote 1,6-difunctionalization of alkenes under visible-light irradiation to efficiently create C(sp^3^)-O and C(sp^3^)-C(sp^2^) bonds at the benzylic position in the presence of O- and C-nucleophiles, respectively (Figure 57) [96]. The reaction of alkenes with alkyl radical precursors and O-/C-based nucleophiles under mild photoredox catalysis gave 1,6-difunctionalized products **62**/**63** in fair to good yields. A proposed mechanism suggests that varieties of C-centered radicals generated from 2-bromo-2,2-difluoro(or 2-monofluoro)acetates, amides, and alkyl and aryl 2-bromoacetates under redox neutral conditions, reacting with the alkenes to afford vicinal radical intermediates **M-115**. The 1,5-HAT of **M-115** produces a distant benzylic radical **M-116** followed by SET oxidation and nucleophilic trapping with O-/C-based nucleophiles to furnish the desired products.

In 2020, the Wang group reported a 1,6-azidotrifluoromethylation reaction of alkenes. The Fe-catalyzed reaction of alkenes, Togni’s reagent II and TMSN_3_ gave difunctionalized products **64** in moderate to excellent yields (Figure 58) [97]. A reaction mechanism suggests that the CF_3_ radical derived from the Togni II reagent adds to the terminal carbon of alkenes. The resulting radicals **M-117** undergo 1,5-HAT to form **M-118** which then react with Fe-N_3_ complex to give products **64a**–**b**. This reaction could be extended for 1,7-bifunctionalized via the 1,6-HAT to afford the corresponding products **64c**–**e**.

A method for visible-light-induced 1,6-oxyfluoroalkylation of alkenes was introduced by the Ma group in 2019 [98]. It has a unique reaction sequence of addition of Rf radical to alkene followed by 1,5-HAT and Kornblum oxidation with DMSO to give products **65** (Figure 59).

In 2021, Chen and colleagues reported a photoredox 1,6-difunctionalization reaction of azaaryl-attached alkenes. The reaction of azaaryl alkenes and RfSO_2_Na in the presence of photosensitizer dicyanopyrazine (DPZ) afforded 1,6-deuteroalkylation products **66** in good yields (Figure 60) [99]. Some commercially available fluoroalkanesulfinic acid sodium salts can smoothly undergo single-electron oxidation to generate fluoroalkyl radicals mediated by visible light. Then, the radical addition of unactivated terminal olefins with the fluoroalkyl radical generates the carbon radical **M-119**, and the 1,n-HAT process is carried out to afford the key radical **M-120**. The relative anion **M-121** generated by the reduction of **M-120** undergoes deuteration with D_2_O to deliver the final 1,6- or 1,7-bifunctionalized product **66**.

Yu and colleagues, in 2020, introduced a photoredox reaction for the difunctionalization of alkenes with CO_2_ and CF_3_ groups. The CF_3_ radical generated from CF_3_SO_2_Na adds to the alkenes followed by 1,5-HAT to afford radicals **M-122** which are then reduced by Ir^II^ to anions **M-123**. Nucleophilic reaction of **M-123** with CO_2_ gives product **67** after protonation (Figure 61) [100]. Other than CO_2_, electrophiles such as aryl aldehydes, aromatic ketoesters and benzyl bromides can be used for making diverse difunctionalized products. In 2022, the Yu group reported photoredox 1,6- and 1,7-dicarboxylation reactions of alkenes with CO_2_. A variety of unactivated aliphatic alkenes can undergo double carboxylations to afford dicarboxylic acids **68** in moderate to good yields (Figure 62) [101].

In 2016, the Zhu group presented a new method for the synthesis of *ε*-CF_3_-substituted amides involving the 1,5-HAT to form acyl radicals as the key step [102]. The Cu-catalyzed reaction of alkenals, Togni’s reagent II and amines in the presence of CuSO_4_ and K_2_CO_3_ afforded products **69** in good yields (Figure 63). In the reaction process, CF_3_ radical derived from Togni II adds to the terminal carbon of alkenes followed by 1,5-HAT, trapping the acyl radicals **M-124** with amines, oxidation via SET afford products **69** after deprotonation.

The Zhu group, in 2017, reported another 1,6-difunctionalization reaction involving the remote-HAT process to form acyl radicals. The Pd-catalyzed reaction of alkenyl aldehydes, arylboronic acids and fluoroalkyl bromides afforded difluoroalkylated ketones **70** in good to excellent yields (Figure 64) [103]. A reaction mechanism suggests that the fluoroalkyl radicals generated from fluoroalkyl halides add to the alkene moiety of alkenyl aldehydes, followed by 1,5-HAT to form the acyl radicals **M-125**, transmetallation with Pd^I^ species and then with ArB(OH)_2_ to afford aryldifluoroalkylation products **70** after reductive elimination of the Pd-cat. The Zhu group also reported a similar reaction of alkenyl aldehydes, arylboronic acids and tertiary *α*-carbonyl alkyl bromides under Ni-catalysis to afford quaternary carbon-containing ketones **71** (Figure 65) [104].

In 2017, the Gagosz group reported a Cu-catalyzed remote oxidative difunctionalization reaction of alkenols. The reaction of alkenols and Togni’s reagent II in the presence of Cu(OAc)_2_ and bipyridine afforded various trifluoromethylated ketones **72** in good yields (Figure 66) [105]. In the reaction process, the CF_3_ radical derived from the Togni II reagent adds to alkenes followed by 1,5- or 1,6-HAT to afford more stable *α*-hydroxy radicals **M-126** which are then oxidized by Cu^II^ to provide 1,6- or 1,7-bifunctionalized product **72**. The Liu and Luo groups also reported these types of reactions [106,107]. In 2018, Liu and colleagues disclosed a reaction of alkenols to introduce sulfonyl, phosphony-, and malonate groups to the products **73**–**75** (Figure 67) [108].

The radical-induced 1,2-migration of boronate complexes has been recently developed for making functionalized organoboronic acid esters [109,110]. In 2021, the Studer group reported a photo reaction of alkenyl boronate complexes with a cascade sequence of perfluoroalkyl radical addition, 1,5- or 1,6-HAT, SET oxidation, and 1,2-alkyl/aryl migration for the construction of remotely 1,5- and 1,6-difunctionalized organoboronic esters (Figure 68) [111]. The alkenyl boronate can be produced in situ by the reaction of the related alkenyl boronic esters with alkyl/aryl lithium reagents. By changing the alkyl/aryl lithium donors and perfluoroalkyl radical precursors, a wide variety of highly functionalized organoboronic esters **76** and **77** can be produced.

Recently, Xia and colleagues reported a 1,6-iminosulfonylation reaction by reacting alkenes with oxime esters to afford diverse imine sulfones **78** in moderate yields (Figure 69) [112]. In the reaction process, an iminyl radical and a sulfonyl radical are generated from the benzophenone oxime ester via homolysis of the N−O bond under photocatalytic conditions. The sulfonyl radical adds to the alkenes followed by 1,5-HAT to afford key radicals **M-127**. The coupling of **M-127** and the iminyl radical gives products **78**.

## 6. Conclusions

Remote radical difunctionalization presents a new research field that is currently the subject of much interest. Most papers summarized in this article have been published within the last five years. Among the different kinds of remote difunctionalization reactions, 1,4- and 1,6-difunctionalizations have been well established due to the development of suitable substrates such as 1,3-dienes/1,3-enynes and 6-subsitituted alk-1-enes. For future work to extend the scope of difunctionalization reactions, design and development of new substrates that bear the scaffolds suitable for expected radical rearrangements *via* resonance, hydrogen atom/group transfer, and strained ring opening are the key factors for success. The recent developments in photoredox reactions, electrochemical reactions, and transition metal-catalyzed coupling reactions provide new avenues for conducting the initial radical reactions as well as the second functionalization reactions. Many newly developed reagents, such as Togni’s, could be utilized to incorporate CF_3_ and other groups into products with medicinal chemistry and drug development applications. We have no doubt that synthetically efficient, operationally simple, and functional-group-diversified remote radical difunctionalization reactions will enjoy more fruitful years to come. We hope that the chemistry highlighted in this paper can be helpful for those who wish to better understand the current status and want to make contributions to the field.

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
