# Peer review of "Remote Radical 1,3-, 1,4-, 1,5-, 1,6- and 1,7-Difunctionalization Reactions†"

_molecules, 2023, doi:10.3390/molecules28073027_

Round 1

Reviewer 1 Report

This review manuscript deals with the most recent achievements in radical remote difunctionalization reactions. It is a complete and comprehensive work, which fits within the scope of this journal.

It is well-structured, and schemes are very clear. It would be nice to mention in the introduction the scope of the review, i.e. be more precise than "recent developments" (p.1, l.37). The reviewer finds that the section on trifunctionalization reactions is out of the scope of this review and should be removed. The manuscript is in general well written, although sentences, especially when mechanisms are described should be more punctuated to avoid long and "heavy" sentences and thus problems of comprehension (for examples see p.3, l.78-82 or p.25, l.464-468 or p.35, l.649-655). While mechanisms are described for each reaction, the reviewer is missing comments on the scope of the reactions and for mechanisms, role of metals is sometimes omitted (for example p.27, l.500-504, photoredox not mentioned at all). Please revise the comments below.

- Scheme 4, it is not obvious at first sight to see where the red oxygen does come from? Show in the reactant what is XO-.

-Scheme 8, it would probably be more interesting to discuss the presence of Cu or not in the two reactions rather than just focus on TMSCN vs DMSO.

-Scheme 12, please comment the change of oxidation of the palladium from M-27 to M-28. (same for scheme 14).

- Sheme 21, p.13, l.258: it is said "in the presence of Et3N", yet in the scheme Et3N is written to be involved in a second step. Please precise. In the scheme, check number 20. Is the role of the ligand on copper discussed in the paper to explaine the difference of reactivity? And why is the reaction in scheme 22 fully selective to 1,4-functionalization?

- p.15, l.298: reductive elimination is always from the oxidized form of the metal. It is incorrect to say "reductive elimination of LCu(I)Br". You should say "reductive elimination of LCu(III)Br". (same p.16, l.313; p.19, l.373; p.23, l.423).

- p.16, l.324: please precise/rephrase "via the chloride advanced oxidation process"

- p.20, l.386: "reductive elimination of NHC", this is not a reductive elimination process, please correct.

- Scheme 36, which specie is M-78? And difference between path a and b is not clear.

- p.24, l.446, precise what is DHP esters.

- Scheme 44, what is CF2G?

- p.39, l.714: "coupling with Pd(I) species and then with ArB(OH)2" This is not a coupling but a transmetallation step.

Typos, missing words have been spotted and are listed below. The authors are encouraged to check the whole manuscript.

- p.1, l.15: "introduction of functional" instead of "introduction functional" (same p.1, l.31)

- In each title of section "difunctionalization" instead of "difunctionalizaiton"

- p.3, l.87: "difluorinated" instead of "diflourinated" (same p.4, l.91)

- p.6, l.120: "resulting" instead of "resulted"

- p.6, l.129: "coupling for " instead of "coupling of for"

- Scheme 9, write the counter anion PF6 in the name of the Ir catalyst to be more consistent.

- Problems of sizing of the text p.7, l.146 and 149.

- Be careful with tense consistency in the description of mechanisms. Example in p.6, l.131-134: promote, add, afford, gave.

- p.10, l.203: "terminal position of the olefin" instead of " olefin terminus"

- Scheme 15, if R-M corresponds to R2Zn, then please replace it. "reaction" instead of "rection" in the title.

- p.11, l.216 and 217: "oxidized" instead of "oxidative", "gives" instead of "give"

-p.11, l.221: "allenyl" instead of "allneyl" (same p.19, l.372)

- rephrase p.13, l.250

- p.14, l.282: remove as, and rephrase "the nucleophile for send functionalization"

- p.17, l.330: what do you mean by "with NBS and H2O sing TEMPO"? "to afford" instead of 'to afforded".

- p.18, l. 341: "excited state" instead of "active state" and l. 343-344: "alkyl radical" instead of "radical alkyl radical" and "afforded" instead of "affording".

- p.8, l.356: missing a verb in "which then R3SO3Na to suflonyl"; l.359: "created" instead of "creating"

- p.19, l.366-368: no conjugated verb in the sentence

- p.20, l.389: "carbonylated" instead of "carbonylted"

- p.21, l.406: "complexes" instead of "complexs" and "are resulting" instead of "are resulted"

- p.23, l.422 and 423: "isocyanocopper" instead of "isocynocopper" and "to form" instead of "with to form"

- p.24, l.437-439: correct "interact", "to form generating", "followed to form".

- Scheme 39, no point at equiv

- Scheme 46, check the matching of colors between reagent and product.

- p.30, l.567: what do you mean by "the reaction of VCPs and SF5Cl in alkanes"?

- other spotted typos p.25, l.466; p.27, l.499,501; p.28, l.525, p.29, l.546; p.30, l.566; p.32, l.605; p.33, l.610; p.33, l.625, p34, l.629, p.35, l.643; p.36, l.658; p.37, l.669,678; p.38, l.687,688; p.39, l.715-716; p.40, l.728; p.46, l.819,828

Author Response

Reviewer 1#

This review manuscript deals with the most recent achievements in radical remote difunctionalization reactions. It is a complete and comprehensive work, which fits within the scope of this journal.

It is well-structured, and schemes are very clear. It would be nice to mention in the introduction the scope of the review, i.e. be more precise than "recent developments" (p.1, l.37). The reviewer finds that the section on trifunctionalization reactions is out of the scope of this review and should be removed. The manuscript is in general well written, although sentences, especially when mechanisms are described should be more punctuated to avoid long and "heavy" sentences and thus problems of comprehension (for examples see p.3, l.78-82 or p.25, l.464-468 or p.35, l.649-655). While mechanisms are described for each reaction, the reviewer is missing comments on the scope of the reactions and for mechanisms, role of metals is sometimes omitted (for example p.27, l.500-504, photoredox not mentioned at all). Please revise the comments below.

Answer: Thanks for reviewer’s positive comments and good suggestions.  

  • "recent developments". We revised the content to “Highlighted in this paper are the development on this topic. Most works are published in last five years”.
  • “trifunctionalization reactions”. The section for trifunctionalization reactions is deleted.
  • “heavy sentences”. Those pointed by referee are revised.
  • “the scope of the reactions and for mechanisms”. We do have limited scope discussion (such as for Scheme 11), but didn’t do full discussions on each cases due to many references (100+) are cited in this paper. We hope the readers could get the information from the original papers. For the same reason, no detailed mechanism discussion is given to each case, especially for those which shares the same mechanism with other cases which have already discussed in this paper. In addition to the information in the text, readers could get the mechanism information from the Schemes.
  • For the example of p.27, l.500-504. “photoredox reaction” was mentioned in the text (I.498). More information could be found in Scheme 44.

- Scheme 4, it is not obvious at first sight to see where the red oxygen does come from? Show in the reactant what is XO-.

Answer: XO- could be tBuOO-, updated in the revised MS.

-Scheme 8, it would probably be more interesting to discuss the presence of Cu or not in the two reactions rather than just focus on TMSCN vs DMSO.

Answer: done

-Scheme 12, please comment the change of oxidation of the palladium from M-27 to M-28. (same for scheme 14).

Answer: done

- Sheme 21, p.13, l.258: it is said "in the presence of Et3N", yet in the scheme Et3N is written to be involved in a second step. Please precise. In the scheme, check number 20. Is the role of the ligand on copper discussed in the paper to explaine the difference of reactivity? And why is the reaction in scheme 22 fully selective to 1,4-functionalization?

Answer: The 1,2- vs 1,4- addition is controlled by the ligand L1 or L2.

- p.15, l.298: reductive elimination is always from the oxidized form of the metal. It is incorrect to say "reductive elimination of LCu(I)Br". You should say "reductive elimination of LCu(III)Br". (same p.16, l.313; p.19, l.373; p.23, l.423).

Answer: fixed

- p.16, l.324: please precise/rephrase "via the chloride advanced oxidation process"

Answer: changed to “via the HAT with chloride radical”

- p.20, l.386: "reductive elimination of NHC", this is not a reductive elimination process, please correct.

Answer: Changed to “elimination of NHC”.

- Scheme 36, which specie is M-78? And difference between path a and b is not clear.

Answer: M-78 is LCuIIAr as shown in the Scheme (above path a). The difference is path b form M-80.

- p.24, l.446, precise what is DHP esters.

Answer: Indicated the DHP esters in Scheme 40.

- Scheme 44, what is CF2G?

Answer: It should be “CF2CO2R”.

- p.39, l.714: "coupling with Pd(I) species and then with ArB(OH)2" This is not a coupling but a transmetallation step.

Answer: fixed

- Typos, missing words have been spotted and are listed below. The authors are encouraged to check the whole manuscript.

- p.1, l.15: "introduction of functional" instead of "introduction functional" (same p.1, l.31)

- In each title of section "difunctionalization" instead of "difunctionalizaiton"

- p.3, l.87: "difluorinated" instead of "diflourinated" (same p.4, l.91)

- p.6, l.120: "resulting" instead of "resulted"

- p.6, l.129: "coupling for " instead of "coupling of for"

- Scheme 9, write the counter anion PF6 in the name of the Ir catalyst to be more consistent.

- Problems of sizing of the text p.7, l.146 and 149.

- Be careful with tense consistency in the description of mechanisms. Example in p.6, l.131-134: promote, add, afford, gave.

- p.10, l.203: "terminal position of the olefin" instead of " olefin terminus"

- Scheme 15, if R-M corresponds to R2Zn, then please replace it. "reaction" instead of "rection" in the title.

- p.11, l.216 and 217: "oxidized" instead of "oxidative", "gives" instead of "give"

-p.11, l.221: "allenyl" instead of "allneyl" (same p.19, l.372)

- rephrase p.13, l.250

- p.14, l.282: remove as, and rephrase "the nucleophile for send functionalization"

- p.17, l.330: what do you mean by "with NBS and H2O sing TEMPO"? "to afford" instead of 'to afforded".

- p.18, l. 341: "excited state" instead of "active state" and l. 343-344: "alkyl radical" instead of "radical alkyl radical" and "afforded" instead of "affording".

- p.8, l.356: missing a verb in "which then R3SO3Na to suflonyl"; l.359: "created" instead of "creating"

- p.19, l.366-368: no conjugated verb in the sentence

- p.20, l.389: "carbonylated" instead of "carbonylted"

- p.21, l.406: "complexes" instead of "complexs" and "are resulting" instead of "are resulted"

- p.23, l.422 and 423: "isocyanocopper" instead of "isocynocopper" and "to form" instead of "with to form"

- p.24, l.437-439: correct "interact", "to form generating", "followed to form".

- Scheme 39, no point at equiv

- Scheme 46, check the matching of colors between reagent and product.

- p.30, l.567: what do you mean by "the reaction of VCPs and SF5Cl in alkanes"?

- other spotted typos p.25, l.466; p.27, l.499,501; p.28, l.525, p.29, l.546; p.30, l.566; p.32, l.605; p.33, l.610; p.33, l.625, p34, l.629, p.35, l.643; p.36, l.658; p.37, l.669,678; p.38, l.687,688; p.39, l.715-716; p.40, l.728; p.46, l.819,828

Answer: We really appreciate referee’s careful proofread. All the typos and missing words are fixed.

Reviewer 2 Report

The manuscript titled "Remote Radical 1,3-, 1,4-, 1,5-, 1,6- and 1,7-Difunctionalization Reactions" by Zhang et al. is a nicely written and well addressed review article. I am pleased to accept it for publication in "Molecules". Some minor mistakes, like the proper positioning of schemes and figures in the middle of the page, are required. Mention the temperature of the reactions reported in Scheme 21. Therefore, I am recommending this manuscript for publication.

Author Response

Reviewer 2#

The manuscript titled "Remote Radical 1,3-, 1,4-, 1,5-, 1,6- and 1,7-Difunctionalization Reactions" by Zhang et al. is a nicely written and well addressed review article. I am pleased to accept it for publication in "Molecules". Some minor mistakes, like the proper positioning of schemes and figures in the middle of the page, are required. Mention the temperature of the reactions reported in Scheme 21. Therefore, I am recommending this manuscript for publication.

Answer: Thanks for the comments. The Scheme 21 is updated.

Reviewer 3 Report

The manuscript Ma et al. is an example of an excellent and comprehensive review of the radical 1,3-1,7 difunctionalization reactions. The review is conveniently divided into chapters; the schemes are informative and give an idea of the reaction conditions. The novelty and relevance of the review are beyond doubt. The work is well organized and will be interesting and useful for readership. English language and style are fine.

In my opinion, this manuscript should be accepted in present form.

Author Response

Reviewer 3#

The manuscript Ma et al. is an example of an excellent and comprehensive review of the radical 1,3-1,7 difunctionalization reactions. The review is conveniently divided into chapters; the schemes are informative and give an idea of the reaction conditions. The novelty and relevance of the review are beyond doubt. The work is well organized and will be interesting and useful for readership. English language and style are fine.

In my opinion, this manuscript should be accepted in present form.

Answer: Thanks!